# The Structural Characteristics of an Acidic Water-Soluble Polysaccharide from *Bupleurum chinense DC* and Its In Vivo Anti-Tumor Activity on H22 Tumor-Bearing Mice

**DOI:** 10.3390/polym14061119

**Published:** 2022-03-11

**Authors:** Shuyuan Shi, Mengli Chang, Huiping Liu, Suyun Ding, Zhiqian Yan, Kai Si, Tingting Gong

**Affiliations:** State Key Laboratory of Food Nutrition and Safety, College of Food Science and Engineering, Tianjin University of Science & Technology, Tianjin 300457, China; ssy08300929@163.com (S.S.); 18337341898@163.com (M.C.); dingsy1996@163.com (S.D.); y_zhiqian@163.com (Z.Y.); 18235447391@163.com (K.S.); gong20200703@163.com (T.G.)

**Keywords:** polysaccharide, *Bupleurum chinense DC*, characterization, anti-tumor activity

## Abstract

This study explored the preliminary structural characteristics and in vivo anti-tumor activity of an acidic water-soluble polysaccharide (BCP) separated purified from *Bupleurum chinense DC* root. The preliminary structural characterization of BCP was established using UV, HPGPC, FT-IR, IC, NMR, SEM, and Congo red. The results showed BCP as an acidic polysaccharide with an average molecular weight of 2.01 × 10^3^ kDa. Furthermore, we showed that BCP consists of rhamnose, arabinose, galactose, glucose, and galacturonic acid (with a molar ratio of 0.063:0.788:0.841:1:0.196) in both α- and β-type configurations. Using the H22 tumor-bearing mouse model, we assessed the anti-tumor activity of BCP in vivo. The results revealed the inhibitory effects of BCP on H22 tumor growth and the protective actions against tissue damage of thymus and spleen in mice. In addition, the JC-1 FITC-AnnexinV/PI staining and cell cycle analysis have collectively shown that BCP is sufficient to induce apoptosis and of H22 hepatocarcinoma cells in a dose-dependent manner. The inhibitory effect of BCP on tumor growth was likely attributable to the S phase arrest. Overall, our study presented significant anti-liver cancer profiles of BCP and its promising therapeutic potential as a safe and effective anti-tumor natural agent.

## 1. Introduction

Liver cancer is the sixth highest-occurring primary cancer and the third major cause of cancer-related death (8.3%), second only to lung cancer (11.4%) and colorectal cancer (10%) [1]. Liver cancer can be divided into three categories, of which the most prominent is hepatocellular carcinoma, accounting for up to 75% to 85% [2]. Early liver cancer is easy to metastasize and difficult to diagnose. Besides, anti-cancer treatment commonly causes toxic side effects to healthy organs as well as immune systems. Moreover, the treatment cost is relatively high [3]. These problems ultimately led to the low cure rate of liver cancer. Patients with advanced liver cancer generally opt for surgical resection combined with other chemotherapeutic treatments. However, the surgical risk is high, while a cure is rare [4]. As the number of patients with liver cancer continuously increases, researchers have committed to improving liver cancer treatment by exploring effective natural anti-liver cancer agents with low toxicity.

Over the past few decades, the natural polysaccharide has attracted more attention because of its multiple health care benefits. Polysaccharides are widely distributed in nature and are implicated in different disease states, pathological processes, and aging [5]. Studies have shown that plant polysaccharides from various sources present varied specificity and a large diversity in their chemical structures [6]. Moreover, the biological activities of polysaccharides have been shown to be highly dependent on the chemical structure [7]. Exploring the therapeutic actions of polysaccharides, promoting them as potential drugs, transforming and synthesizing them has become a hot spot in the fields of life sciences and medicine.

*Bupleurum chinense DC* belongs to the *Bupleurum* spp., a plant of the *Umbelliferae* family [8]. The medicinal part is the dried root of *Bupleurum chinense DC*. As a famous traditional Chinese medicine, *Bupleurum chinense DC* is often used to treat diseases such as antipyretics, labor pains, and colds [9]. *Bupleurum chinense DC* polysaccharide is an essential active component of *Bupleurum chinense DC*, the use of which has been widely reported for the intervention of different disease states and pathological processes, i.e., slow down of inflamm-aging, alleviation of kidney injury in diabetic patients, reduction of oxidation, improvement of the sepsis prognosis, regulation of macrophage functions, inhibition of sarcoma S180 growth in tumor-bearing mice, treatment for melanoma, reduction of inflammation and liver protection, reduction of lipopolysaccharide (LPS) induced acute lung injury in mice, action against chronic gastric ulcers, etc. [10,11,12,13,14,15,16,17,18,19].

In this study, an acidic water-soluble *Bupleurum chinense DC* polysaccharide (BCP) was extracted and purified from the root of *Bupleurum chinense DC*. The basic structure of BCP was characterized and its in vivo anti-tumor activity was evaluated using the H22 tumor-bearing mice models. Overall, our research discovered *Bupleurum chinense DC* as a potential natural therapeutic agent for safe and effective liver cancer treatment.

## 2. Materials and Methods

### 2.1. Materials and Reagents

*Bupleurum chinense DC* roots, produced in Dingxi (place in Gansu), harvested in late September, were dried, crushed, and passed through an 80 mesh sieve to obtain *Bupleurum chinense DC* powder, which was placed in a desiccator and sealed for storage (relative humidity was 70 ± 5%, the temperature was 25 ± 5 °C). Standards of monosaccharides and 5-fluorouracil were purchased from Sigma (St. Louis, MO, USA). T-series dextran was acquired from Solarbio (Beijing, China). Sephadex G-150 was purchased from Shanghai Yuanye Biological Technology Co., Ltd. (Shanghai, China). A JC-1 mitochondrial membrane potential detection kit was gained from Solarbio Science & Technology Co., Ltd. (Beijing, China). Annexin V-FITC/PI apoptosis detection kit was purchased from Dalian Meilun Biotechnology Co., Ltd. (Dalian, China). The cell cycle analysis and apoptosis kit were obtained from Beyotime Biotech (Nantong, China). All other chemical reagents and materials were of analytical grade.

### 2.2. Preparation of BCP

Before preparing our polysaccharide, we optimized the extraction process of polysaccharide. As shown in Appendix A, single-factor experiment was conducted with liquid-to-material ratio, extraction temperature, extraction time, and extraction times as the main factors. Based on the results of preliminary experiment, Box-Behnken (Appendix A) was designed with independent three variables (A, liquid-to-material ratio; B, extraction temperature; C, material extraction time) to obtain the optimal extraction process of crude polysaccharides of *Bupleurum chinense DC* (CBCP). The significance coefficients of the model were evaluated by ANOVA (Appendix A). The response surface and contour plots for the effects of different parameters on CBCP yields are shown in Appendix A.

According to the parameter results of the above optimization process, the preparation process of BCP was shown in Figure 1. First of all, crude polysaccharides were extracted using the ethanol subsiding method [20]. Dried *Bupleurum chinense DC* powder (20 g) was filtered through an 80-mesh sieve and dissolved in distilled water with a solid to liquid ratio of 1:20. It underwent direct-heating extraction in a thermostat water bath at 90 °C for three hours and then centrifuged for 15 min with a high-speed centrifuge at a speed of 8000 r/min. The process was repeatedly extracted three times, and the supernatant was combined to obtain the supernatant. The resulting supernatant was condensed to one-third of its original volume by a rotary evaporator. Next, it was mixed with 1.5 times the volume of 95% ethanol and was stored at 4 °C overnight. Then we collected the precipitation by centrifugation and allowed the ethanol to be evaporated in a ventilated place. After the ethanol evaporation, a small amount of distilled water was added to dissolve it, and the impurities were removed by centrifuging at a speed of 8000 r/min. Four volumes of Sevag solution were put into the extract, followed by 20-min shaking, and then the solution was placed in a separatory funnel. After the mixture was layered, the upper layer of polysaccharide solution was collected, and the step was repeated 8 times to eliminate protein impurities [21]. The collected polysaccharide solution was then put on a rotary evaporator to remove any residual Sevag solution. Afterwards, the CBCP could be obtained after lyophilization.

Then, the isolation and purification of CBCP were conducted using the column chromatography method [22]. We fully dissolved it in distilled water, followed by dialysis (10 kDa) with flowing and distilled water alternately for three days. The polysaccharides extracts were then freeze-dried and re-dissolved in deionized water to achieve final concentration of 20 mg/mL. Then, the purification step was performed using Sephadex G-150 gel column (1.6 × 60 cm). The eluent was in distilled water, and the flow rate was set at 0.1 mL/min. Automatic fraction collector was used to collect one tube every 15 min. The phenol-sulfuric acid method was used to detect the polysaccharide content of the elution peak. Next, the main peak part was collected and lyophilized to obtain the pure *Bupleurum chinense DC* polysaccharide (BCP) [23].

### 2.3. BCP Characterization

#### 2.3.1. Chemical Components Analysis

The total carbohydrate and reducing carbohydrate content in BCP were measured with phenol-sulfuric acid and 3, 5-dinitrosalicylic acid (DNS) method with D-glucose as the standard [24]. Using the bovine serum albumin standard, the protein content in BCP was measured with the Coomassie brilliant blue (G-250) method [25]. The content of uronic acid in BCP was determined by the carbazole-sulfuric acid method with galacturonic acid as the standard [26].

#### 2.3.2. UV Spectroscopy Analysis 

BCP solution of 1 mg/mL was prepared in distilled water. Using UV spectrophotometry (UV-25900PC, Japan), the absorption was measured by a full wavelength scan in the 190–500 nm wavelength range [27]. The quantification for nucleic acid and protein contents was determined based on the absorption at 260 nm and 280 nm wavelength, respectively [28]. 

#### 2.3.3. Molecular Weight Analysis by HPGP

The effects of different extraction temperature (70 °C, 90 °C) and ethanol concentration (60%, 70%, 80%) on the molecular weight of *Bupleurum chinense DC* polysaccharide were compared. Six samples were prepared for analysis [29]. The average molecular weight of BCP was evaluated by HPGPC (Agilent Technologies, Palo Alto, CA, USA). The sample was prepared by weighing 1 mg BCP, dissolving in 2 mL ultra-pure water, and passing through a 0.22 μm semipermeable membrane. T-series dextrans T-10 (1 × 10^4^ Da), T-40 (4 × 10^4^ Da), T-70 (7 × 10^4^ Da), T-110 (1.1 × 10^5^ Da), T-500 (5 × 10^5^ Da), and T-2000 (2 × 10^6^ Da) were used as standard products [30]. The experimental conditions and the specific parameters of the instrument were as follows: Aglient 1200 high-performance liquid chromatograph (Agilent Technologies, Palo Alto, CA, USA), TSK-gel G4000PWxl column (Agilent Technologies, Palo Alto, CA, USA), Refractive Index Detector (RID) (Agilent Technologies, Palo Alto, CA, USA), ultra-pure water as mobile phase, the flow rate was 0.6 mL/min, the column temperature was 30 °C, the detector temperature was 35 °C, and the injection volume was 20 μL. The establishment of the standard curve by the T series standard products and the average molecular weight of BCP counted with the standard curve.

#### 2.3.4. FT-IR Spectrum Analysis

The sample of 1 mg BCP with dried potassium bromide powder of 150 mg was weighed, ground thoroughly in a bowl, poured into a mold, and pressed into a transparent sheet. It was scanned using FT-IR spectrometer (Bruker VECTOR-22, Karlsruhe, Germany), and the recorded infrared spectrum was at the range of 400–4000 cm^−1^ [31].

#### 2.3.5. Monosaccharide Composition Analysis by IC

Firstly, BCP of 5 mg was accurately weighed and added to 2 mol/L trifluoroacetic acid (TFA). After filling the tube with N_2_, it was hydrolyzed in an oil bath at 115 °C for three hours. Then the TFA was removed. The methanol solution was added to remove the excess TFA and then dried three times with N_2_. Finally, all TFA was removed to obtain the final degradation product. The obtained hydrolysate was dissolved in ultra-pure water and compounded into a concentration of 100 ppm solution [32]. Nine monosaccharides (rhamnose, arabinose, fucose, galactose, glucose, xylose, mannose, glucuronic acid, and galacturonic acid) were compounded into a mixed standard with a concentration of 100 ppm. The monosaccharide composition of BCP was detected by ion chromatography (IC). The detector was Dionex ICS2500; the analytical column was PA20 (150 mm × 3 mm, the column temperature: 30 °C), the eluents were NaAc solution (100 mm) and NaOH solution (6 mm), the injection volume was 1 mL, and the flow rate was 0.45 mL/min [33].

#### 2.3.6. NMR Spectroscopy Analysis

The 60 mg BCP was placed in the NMR tube and dissolved in 0.5 mL D_2_O. The ^1^H NMR and ^13^C NMR spectra were recorded at 25 °C with a Bruker AMX-500 NMR.

#### 2.3.7. Scanning Electron Microscopy Analysis

The microstructure of BCP was observed by using scanning electron microscopy (SEM). The dried BCP powder was fixed on the wall by double-sided adhesive tape, and the excess floating powder was blown off by a rubber suction bulb, then sputtered with a layer of gold for the sake of conduction [34]. Experimental conditions were configured as SEM (SU1510, Hitachi, Japan), with a magnification of 100×, 1000×, and 3500×, respectively.

#### 2.3.8. Thermal Analysis by TGA and DSC

The thermal properties of BCP were analyzed by TGA (Perkin–Elmer, Waltham, Massachusetts, USA) and DSC (Q2000, TA, New Castle, PA, USA). In the TGA test, a 10 mg of sample was placed in an aluminum pan and heated from 25 to 600 °C at a rate of 10 °C/min in N_2_ atmosphere [35]. In the DSC test, 10 mg of sample was heated from 25 to 200 °C at a rate of 10 °C/min under N_2_ atmosphere [36].

#### 2.3.9. Congo Red Analysis

The sample of 0.5 mg/mL BCP and 50 mol/L Congo red was prepared in advance. The two reagents mixed with 1 mol/L NaOH solution to adjust the concentration of NaOH [37]. After mixing, the two sets of NaOH solutions with concentrations of 0, 0.1, 0.2, 0.3, 0.4, and 0.5 mol/L were obtained [38]. The derivatization reaction was carried out at 25 °C for 10 min, and the full wavelength scanning was carried out in the 400–600 nm wavelength range with an ultraviolet visible spectrophotometer (UV-Vis).

### 2.4. BCP Anti-Tumor Activity on H22 Tumor-Bearing Mice

#### 2.4.1. The Materials and Conditions of Animals Experimental 

H22 hepatoma cells were purchased from Shanghai Institute of Biological Sciences, Chinese Academy of Sciences. They were stored in DEAE medium supplemented with 10% heat-inactivated fetal calf serum (FCS), 100 U/mL penicillin, and 100 g/mL streptomycin. Next, they were cultured and passaged at 37 °C in 5% CO_2_. Kunming mice (age: 7–8 weeks old, gender: female, body weight: 20 ± 2 g) were provided by SPF Biotechnology Co., Ltd. (Beijing, China). The license code is SCXK 2019-0010. Mice feeding conditions were as follows: relative humidity was 50 ± 5%, the temperature was 22 ± 2 °C, the light-dark cycle was 12 h, and water and feed were provided free of charge. All animal experiment procedures followed the “Regulations on the Management of Laboratory Animals” (China).

#### 2.4.2. Establishment of H22 Tumor-Bearing Mouse Model

After feeding the mice freely for a week, fifty mice were randomly divided into five groups. The groups were as follows:Blank group;Model group;5-Fu group: Injection of 5-fluorouracil as a positive control group (20 mg/kg);LBCP group: Low dose of BCP treatment group (100 mg/kg);HBCP group: High dose of BCP treatment group (300 mg/kg).

The establishment of H22 tumor-bearing mouse model process is shown in Figure 2. Mice in groups 1–5 were given 0.2 mL normal saline intragastric administration at the same time for 7 days. Then, mice in groups 2–5 were inoculated with 10^6^ H22 cells in subcutaneous right forelimb armpit. Group 1 and group 2 were gavaged with 0.2 mL sterile normal saline every day, group 3 was intraperitoneally injected with 20 mg/kg 5-fluorouridine daily, group 4 was gavaged with 100 mg/kg BCP, and group 5 was gavaged with 300 mg/kg BCP (intragastric administration or intraperitoneal injection were made at the same time every day, continuing for 14 days). The weights of mice and their vital signs were recorded during the experiment.

#### 2.4.3. Solid Tumors and Immune Organ Indices

Within 24 h after the last administration, all mice were weighed. After weighing, they were sacrificed by cervical dislocation and then dissected. The solid tumors, thymus, and spleen were removed from all mice. The blood stains were washed with PBS buffer, the surface water was dried with filter paper, and the tissues were weighed. The vernier caliper was used to measure the tumor volume of mice. The tumor inhibition rate (TIR) and immune organ indices were calculated by the following Formulas (1) and (2).
TIR (%) = (the average tumor weight of model group − the average tumor weight of treated group)/the average tumor weight of model group × 100 (1)
Organ index (mg/g) = organ weight (g)/mice wight (g) × 1000 (mg)/1 (g)(2)

#### 2.4.4. FITC-AnnexinV/PI Double Staining Detection

The light-scattering properties of cells changed during apoptosis. PI was used as a fluorescent dye, the tumor tissue was ground and crushed with a cell sieve, and the cell suspension was prepared in PBS. Next, the cell density was diluted to 10^6^ cells/mL. The changes of cell light scattering were measured by flow cytometer to identify apoptotic cells by the test method of FITC-AnnexinV/PI kit.

#### 2.4.5. Cell Cycle Distribution Detection

The cell suspension was prepared as described in Section 2.4.3 and fixed with 70% alcohol overnight at 4 °C. The cell cycle distribution detection is conducted using the manufacturer’s instructions. After washing with PBS to remove ethonal, 50 μL RNase (1 mg/mL) was added. The sample was then digested at room temperature for about 30 min, followed by the addition of 50 μL PI (500 μg/mL propidium iodide) for staining. They were gently mixed and incubated at 4 °C for 10 min, shielded from light. The variation of cell cycles was detected by using a flow cytometer. 

#### 2.4.6. Assay of Mitochondrial Membrane Potential (∆Ψm) 

JC-1 is a fine fluorescence probe that is commonly used to detect mitochondrial membrane potential (ΔΨm). Following the kit instructions, the prepared cell suspension (as described in Section 2.4.3) was blended with the working medium of JC-1 at room temperature and was kept incubated for 30 min. After centrifugation, the precipitates were washed repeatedly with PBS, and then the ΔΨm was measured by flow cytometer.

### 2.5. Statistical Analysis

The experimental findings generated in the experiment are expressed as mean ± standard deviation (X ± SD). The data were analyzed by SPSS software, and ANOVA was used to establish a significant difference. *p* < 0.05 could be considered as statistically significant.

## 3. Results

### 3.1. The Basic Chemical Components and UV-Visible Spectrum Analysis of BCP

The crude polysaccharides of *Bupleurum chinense DC* (CBCP) was obtained from dried *Bupleurum chinense DC* powder by preliminary extraction. Then, the yield of CBCP was determined to be about 5.35%. To purify the product, protein and fat impurities were firstly removed, and small molecule polysaccharides were further eliminated by dialysis procedure. The subsequent purification was conducted using Sephadex G-150 gel columns. The experimental data showed: the total sugar content of BCP was about 93.58 ± 1.34%, the uronic acid content was 9.64 ± 0.35%, and the protein content was 0.65 ± 0.22%. This result preliminarily suggested BCP as an acidic polysaccharide. According to the ultraviolet-visible spectroscopy of BCP (Figure 3), the absence of absorption peaks at 260 nm and 280 nm indicated that BCP contained trace amounts of protein and nucleic acid.

### 3.2. HPGPC and FT-IR Analysis of BCP

As shown in Appendix A, the extraction temperature of *Bupleurum chinense DC* polysaccharide had no obvious effect on its molecular weight. However, with the increase of alcohol concentration, the peak time of HPGPC chromatogram shifted to the right, and its molecular weight became low. 

The HPGPC preprogram of BCP (Figure 4) has shown a single homogeneous narrow peak, indicating that BCP was a homogeneous fraction. According to the standard curve obtained from T series standard products, the regression equation was log Mw = −0.3173 Rt + 8.9584 (R^2^ = 0.9923). We could calculate that the molecular weight of BCP was 2.01 × 10^3^ kDa (Rt: 8.368 min).

The infrared spectrum of BCP (Figure 5) indicates that BCP is a typical polysaccharide [39]. There was a strong and broad band of O-H stretching vibration at 3423.74 cm^−1^, while the absorption peak appeared at 2920.34 cm^−1^ due to C-H stretching vibration. The absorption peaks at 1743.67 cm^−1^ and 1239.21 cm^−1^ were attributed to the existence of uronic acid, which supported the determination of uronic acid described in 2.1 above. The strong absorption peak at 1617.48 cm^−1^ confirmed the characteristic asymmetric stretching of the C=O. Moreover, the absorption peaks at 1439.76 cm^−1^, 1371.18 cm^−1^, and 1331.59 cm^−1^ could be attributed to the variable angle vibration of C-H. The signal peaks at the range of 800–1200 cm^−1^ could be called the carbohydrate fingerprint region, among which the peaks at 1100.46 cm^−1^, 1049.85 cm^−1^, and 1020.46 cm^−1^ were generated by the bending vibration of C-O [40]. The weak absorption peaks at 893.66 cm^−1^ and 831.61 cm^−1^ confirmed that BCP contained both β- and α-type glycosides [41].

### 3.3. Monosaccharide Composition Analysis of BCP

Figure 6A,B showed the ion chromatograms of monosaccharide standards and BCP hydrolysates. The analysis revealed the major compositions of BCP: rhamnose, arabinose, galactose, glucose, and galacturonic acid, with a molar ratio of 0.063:0.788:0.841:1:0.196. In recent years, researchers have isolated different kinds of polysaccharides from *Bupleurum* to explore their structure. The monosaccharide compositions of other *Bupleurum* polysaccharide analogues were compared with BCP, as shown in Appendix A [11,12,15,16,42,43,44,45,46]. The detection of galacturonic acid in BCP was consistent with the previous determination of uronic acid in BCP by the carbazole sulfonic acid method [47].

### 3.4. NMR Results of BCP

In order to further characterize the BCP structure, we performed an NMR analysis. The glycosidic bond with anomeric hydrogen on α-type generated chemical shifts at δ 4.9–5.9 ppm, while β-type caused shift at δ 4.3–4.9 ppm [48,49,50]. As shown in Figure 7A, the chemical shift of ^1^H NMR spectra of δ 3.25–5.30 ppm confirmed BCP as a typical polysaccharide with both α-type glycosidic bond and β-type glycosidic bond [51]. The solvent peak of D_2_O appeared at δ 4.70 ppm. The ^1^H NMR spectra also showed that protons of H2 to H6 had chemical shifts in the range of δ 3.25 to δ 4.19 ppm [52,53]. 

As shown in Figure 7B, there were five signal peaks in the range of δ 99.62–107.44 ppm at anomeric C-1 region in the ^13^C spectrum, which attributed to the α-type glycosidic bond and β-type glycosidic bond in BCP. The presence of uronic acid has been confirmed once again with the chemical shift signal at δ 170.66 ppm. Together with previous studies, our findings suggest: (1) the ^1^H signal at δ 4.99 ppm and the ^13^C signal at δ 107.44 ppm demonstrated the existence of β-D-Gal in BCP [54]; (2) The ^1^H signal at approximately δ 4.38 ppm and the ^13^C signal around δ 107.08 ppm were due to the existence of β-L-Rha. Besides, the ^1^H signal at δ 5.04 ppm and the ^13^C signal at δ 103.40 ppm demonstrated the presence of α-L-Araf [55]. The signals were observed at δ 100.35 ppm of the BCP ^13^C spectrum and δ 5.12 ppm of the BCP ^1^H spectrum could be owing to the existence of α-D-Gal. Furthermore, the ^1^H signal at approximately δ 5.30 ppm and the ^13^C signal signals at δ 99.62 ppm suggested the presence of α-D-Glu in BCP. To conclude, the data above were consistent with the results from FT-IR and monosaccharide composition. The anomeric carbon signals occurred at δ 60.49–83.86 ppm, attributable to the signal of the C-2 to C-6 [56].

### 3.5. The Molecular Morphology of BCP 

The microscopic surface morphology of BCP was studied using the scanning electron microscope with different magnifications: 100× (Figure 8A), 1000× (Figure 8B), and 3500× (Figure 8C). This result indicated that BCP was flaky or clastic with a rough surface and predominantly layered structure, with sizes from 100 to 1300 microns [57].

### 3.6. Thermal Analysis of BCP

As shown in Figure 9A, there were three distinct weight-loss stages in TGA curve of BCP at the range of 25–600 °C. In the first stage, a mass loss of approximately 9.96% occurred near 65 °C, which was assigned to the evaporation of water in the BCP sample [58]. The second mass loss change was occurred at the temperature range of 200–450 °C. At this stage, the most obvious mass loss was about 67.33%, which could be attributed to thermal decomposition polysaccharide [59]. In the last stage, most of the remaining materials were converted into ash and inorganic components at the temperature range of 450–600 °C [60]. The results of TGA showed that BCP had good thermal stability below 200 °C.

The DSC curve (Figure 9B) of the BCP occurred endothermic peak at 67.7 °C, which could be attributed to the evaporation of water in the BCP sample [61]. When the temperature was increased to 200 °C, there was no new endothermic peak that appeared. The results of DSC showed that there was no depolymerization reaction caused by glycosidic bond breaking in the range of 25–200 °C, and the polysaccharide structure was stable at this stage which coincides with the TGA analysis. 

### 3.7. Congo Red Analysis of BCP 

Congo red is an acid dye that forms complexes with polysaccharides with a spatial helical structure. As shown in Figure 10, when the concentration range of NaOH was 0–0.05 mol/L, the absorption wavelength gradually increased until it peaked around 508 nm at 0.05 mol/L. Then, as the concentration of sodium hydroxide increased, the maximum wavelength of absorption decreased and stabilized. The maximum wavelength of the control group was significantly lower than the BCP + Congo group, and a redshift (*p* < 0.05) was observed. These results suggested a triple-helix structure for BCP. Interestingly, the previous literature revealed that the triple-helical polysaccharides could actively induce tumor cell apoptosis [62,63,64,65].

### 3.8. Anti-Tumor Activities In Vivo of BCP on H22-bearing Mice

#### 3.8.1. Weight, Immune Organ Indices, and Tumor Inhibition Rate

The death date and survival rate of mice in each group was shown in Table 1 below. Differences in tolerance among individual mice resulted in the death of mice during the period. However, in general, the survival rate of mice was high, and the data were significant [66].

The weights of mice are shown in Table 2. The mice were similar before inoculation with H22 hepatoma cells (22 ± 1 g), and there was no significant difference between each group. The final weight gain of the model group was markedly different from that in the blank group (*p* < 0.05), suggesting that H22 tumor cells proliferated indefinitely in the model group of mice. The mice in the 5-Fu group had symptoms such as loss of appetite, sluggishness, and depressed mental state. In contrast, the weight of mice in LBCP and HBCP groups was gradually close to that of mice in the blank group. The mental state of mice was also significantly improved, indicating the BCP treatment on H22 tumor-bearing mice showed substantial beneficial effects without severe toxicity.

The tumor inhibition rates calculated by the tumor weights and volumes are listed in Figure 11A. The weight and volume of tumors in the 5-Fu, the LBCP, and the HBCP groups were lower than the model group. To be exact, the tumor inhibition rate (TIR) was 57.53% in the 5-Fu group, 15.44% in the LBCP group, and 37.20% in the HBCP group. The results showed that 5-Fu had the most significant anti-tumor effects, but its side effects on mice were also severe. In contrast, the tumor-inhibiting capacity of BCP increased in a dose-dependent manner without inhibiting the growth state of the mice. The number of immune cells is highly correlated to the weights of immune organs as well as immunological functions [67]. The thymus is an essential lymphatic organ that stores and secretes immune cells and molecules [68]. As the largest immune organ, the spleen is the core of cellular and humoral immunity [69]. The thymus and spleen are the essential immune organs in the body, and their organ indexes can be calculated to evaluate the strength and weakness of immune functions. The organ indexes are listed in Figure 11B. The thymus index of the model group was markedly decreased (*p* < 0.05) compared to the blank group, suggesting that thymus atrophy was accompanied by tumor cell proliferation. The thymus index of the 5-Fu group was the lowest, even lower than that of the model group (*p* < 0.05). The thymus index increased from the LBCP group to the HBCP group compared to the model group, indicating that BCP induced a protective effect on the thymus. The spleens from the model group were remarkably swollen compared to the blank group (*p* < 0.05). Although the spleen indexes in the LBCP and HBCP groups were higher than the 5-Fu group, they were remarkably lower than the model group (*p* < 0.05). The changes in immune organ indexes in the LBCP group and HBCP group illustrated that BCP could enhance the immunity of H22 tumor-bearing mice. These consequences fully suggested that 5-Fu could not only inhibit the rapid proliferation of tumor cells effectively but destroy normal immune organs [70]. Altogether, BCP was shown to protect immune organs while suppressing tumor growth.

#### 3.8.2. Cell Apoptosis Analysis by FITC-AnnexinV/PI

Propidium iodide (PI) is a commonly used red-fluorescent dye that stains the nucleus and chromosomes. Since PI is not permeable to live cells, it can selectively stain dead cells or cells in the middle or late stage of apoptosis [71]. Annexin V staining is another common method to detect apoptotic cells. During apoptosis, the anionic phosphatidylserine (PS) is translocated to the extracellular side of the plasma membrane and binds to Annexin V conjugates with high affinity [72]. Dead and apoptotic cells (in the period of early and late stage of apoptosis) can be distinguished using FITC-AnnexinV/PI staining [73]. As illustrated in Figure 12, the apoptosis rate of the model group without BCP and 5-Fu treatment was 11.32% (early apoptosis rate 9.64%, late apoptosis rate 1.68%). After BCP treatment, the apoptosis rate of the LBCP group was 20.98% (early apoptosis rate 14.60%, late apoptosis rate 6.58%), and the apoptosis rate of HBCP group reached 32.3% (early apoptosis rate 21.30%, late apoptosis rate 12.00%). The apoptosis rates of the LBCP and HBCP groups were remarkably increased compared to the model group (*p* < 0.05). The proportion of early and late apoptosis in solid tumor cells in the 5-Fu group was also remarkably increased compared to the model group (*p* < 0.05). The results have shown that BCP induced apoptosis and inhibited the rapid proliferation of H22 hepatoma cells in a dose-dependent manner.

#### 3.8.3. Cell Cycle Analysis 

The apoptosis-inducing ability of BCP was further explored by PI staining [74]. A complete cell cycle comprises five different phases, including G0, G1, S, G2, and M phases [75]. It was reported that cell cycle arrest at a specific stage would induce tumor cell apoptosis [76]. As shown in Figure 13, compared with the model group, the G0/G1 phase and G2/M phase in the BCP group were significantly decreased (*p* < 0.05). The percentage of S-phase cells increased by BCP treatment in the LBCP group (18.98%) and the HBCP group (38.69%) when compared with the model group (15.97%). Therefore, we speculated that BCP could arrest solid tumor cells in the S phase of the cell cycle and induce tumor cell apoptosis in a dose-dependent manner.

#### 3.8.4. Mitochondrial Membrane Potential (MMP) Analysis 

In recent years, research has shown that the decrease of MMP is highly correlated with the apoptosis of cells under various influencing factors [77]. As shown in Figure 14, the fluorescence intensity of the untreated model group was as high as 94.30%. After BCP treatment, the signal peak gradually moved to the left with the increase of BCP dose, resulting in the fluorescence intensity of 84.30% in the LBCP group and 65.20% in the HBCP group. After 5-Fu treatment, the fluorescence intensity of the 5-Fu group was remarkably increased compared to the model group (*p* < 0.05). The results suggest that BCP might dose-dependently cause the decrease of mitochondrial membrane potential and ultimately lead to apoptosis.

## 4. Discussion

In this paper, we extracted and purified an acidic water-soluble polysaccharide BCP from the root of *Bupleurum chinense DC* and investigated its structure and anti-tumor activity. We showed that BCP was an acid-soluble polysaccharide that contained trace amounts of protein and nucleic acid. As we all know, pharmaceutical polysaccharides have rich biological activities because of their unique monosaccharide composition and special structure. *Bupleurum chinense DC* is a commonly used drug product, which is mainly used to clear away heat and disperse fire [78]. There are also articles mentioning the role of *Bupleurum chinense DC* for soothing and protecting the liver [79]. However, as far as we know, this is the first time that a *Bupleurum chinense DC* polysaccharide with such a large molecular weight has been extracted, and its basic structure and its anti-liver cancer activity in vivo have been explored. The research may contribute to the development of functional food ingredients with the potential to treat liver cancer. The spleen, thymus, and other immune organ indexes can reflect the immune status of the body to a certain extent. After BCP treatment, the body’s immune index was significantly improved compared with 5FU treatment. The results confirmed that drug treatment has irreversible effects on the damage of the body’s immune function, but BCP improved the body’s immune function. It has been reported that mitochondria play an irreplaceable role in cell apoptosis because they can transmit and amplify death signals, and are the central link in the interaction between upstream apoptosis pathway and Caspase pathway and other downstream death pathways [80]. There are three main ways of mitochondria mediated apoptosis. The first is to destroy the antioxidant capacity of cells, the second is to block the production of ATP by breaking the electron chain, and the third is to affect the mitochondrial pathway of cell apoptosis [81]. In this research, BCP induced apoptosis of H22 tumor cells mainly through the third pathway, the cell cycle analysis, FITC-AnnexinV/PI staining, and JC-1 experiments, which have collectively shown that BCP is sufficient to induce apoptosis and inhibit the rapid proliferation of H22 hepatocarcinoma cells in a dose-dependent manner. The inhibitory effect of BCP on tumor growth was likely attributable to the cell cycle arrest (in the S phase) and the activation of mitochondria-related pathways. 

In this paper, we extracted and purified an acidic water-soluble polysaccharide BCP from the root of *Bupleurum chinense DC* and investigated its structure and anti-tumor activity. We showed that BCP was an acid-soluble polysaccharide that contained trace amounts of protein and nucleic acid. The average molecular weight of BCP was 2.01 × 10^3^ kDa, consisting of rhamnose, arabinose, galactose, glucose, and galacturonic acid in the molar ratio of 0.063:0.778:0.841:1:0.196, accompanied by α- and β-type glycosidic residues. On this basis, we also observed its microstructure and found that it has a layered, rough surface with a few fragments. The results of Congo red showed that BCP had a triple-helix structure. Some literatures have shown that the triple helix structure of polysaccharides is a kind of biological macromolecule with special chain structures in nature. It not only has high biological activity, but also has special molecular recognition ability and incomparable functional properties of other polysaccharides [82]. More importantly, we revealed that BCP could induce apoptosis and inhibit the rapid proliferation of H22 hepatoma cells by arresting growing cells in the S phase of the cell cycle in a dose-dependent manner without causing severe toxicity. In conclusion, our paper exhibited promising anti-liver cancer activity of BCP and shed light on the potential use of *Bupleurum chinense DC* as an effective and safe anti-liver cancer treatment.

## Figures and Tables

**Figure 1 polymers-14-01119-f001:**
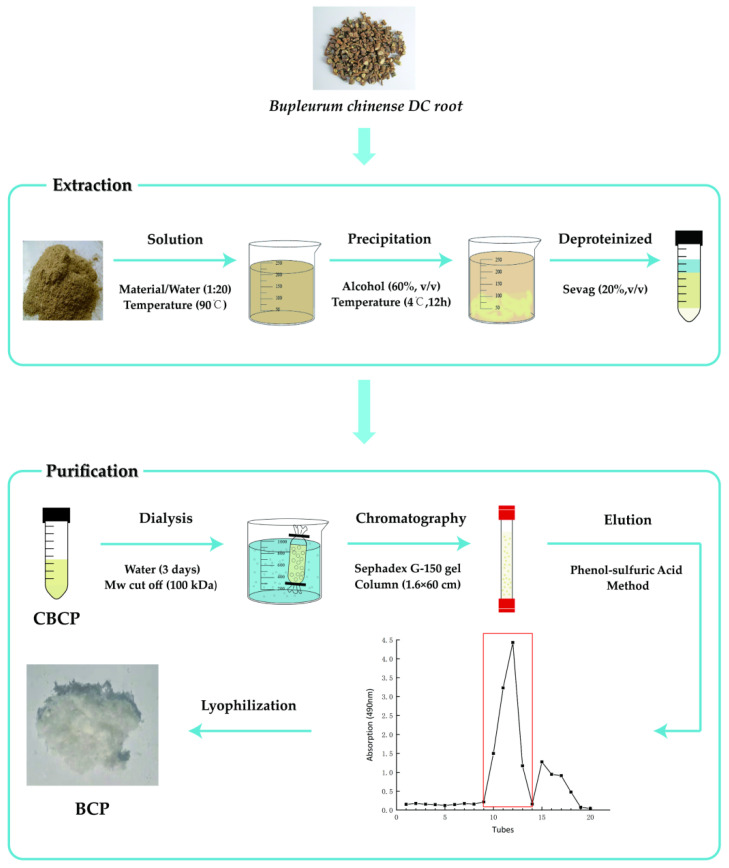
The preparation process diagram of BCP.

**Figure 2 polymers-14-01119-f002:**
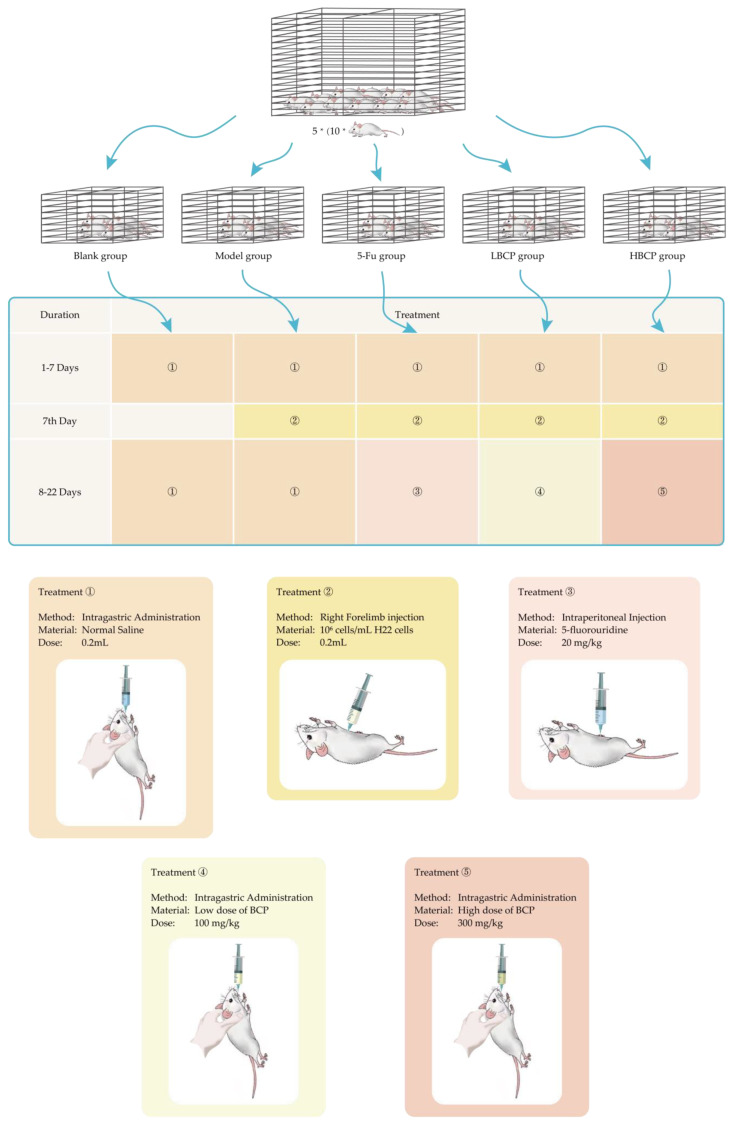
The establishment of H22 tumor-bearing mouse model process.

**Figure 3 polymers-14-01119-f003:**
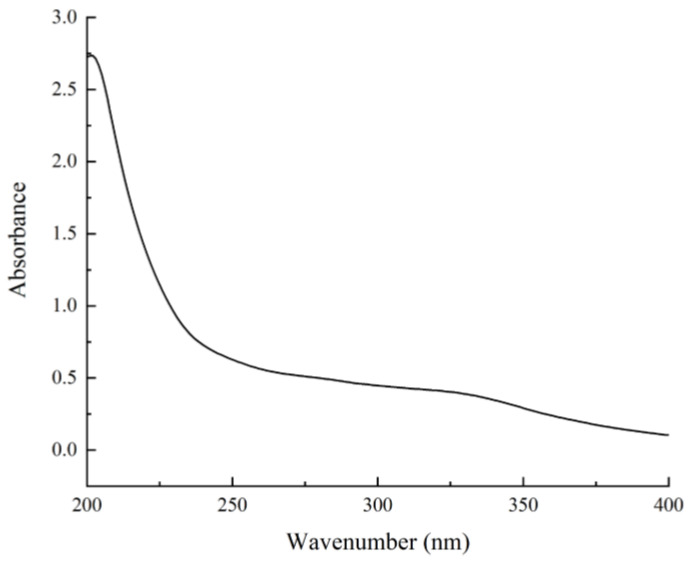
UV-visible spectrum of BCP.

**Figure 4 polymers-14-01119-f004:**
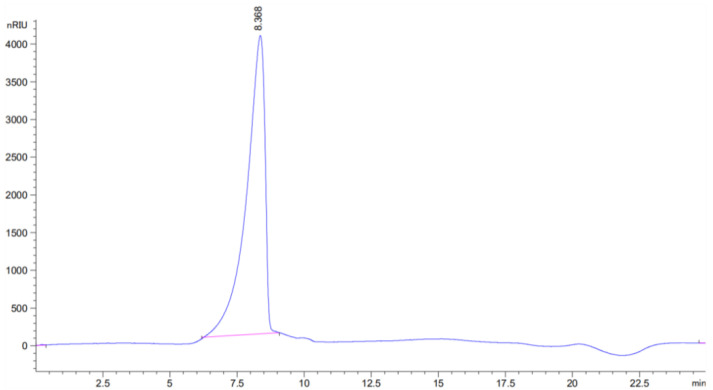
HPGPC chromatogram of BCP.

**Figure 5 polymers-14-01119-f005:**
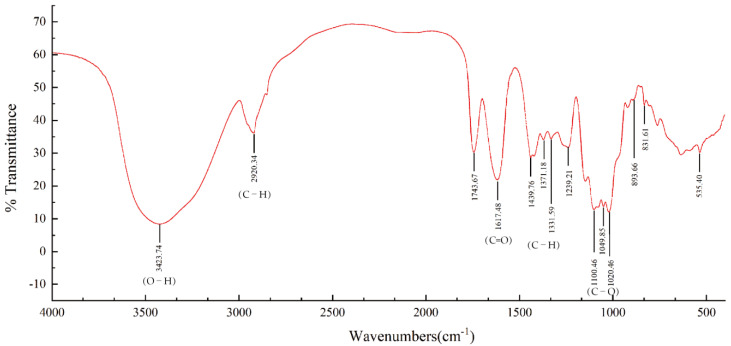
FT-IR spectrum of BCP.

**Figure 6 polymers-14-01119-f006:**
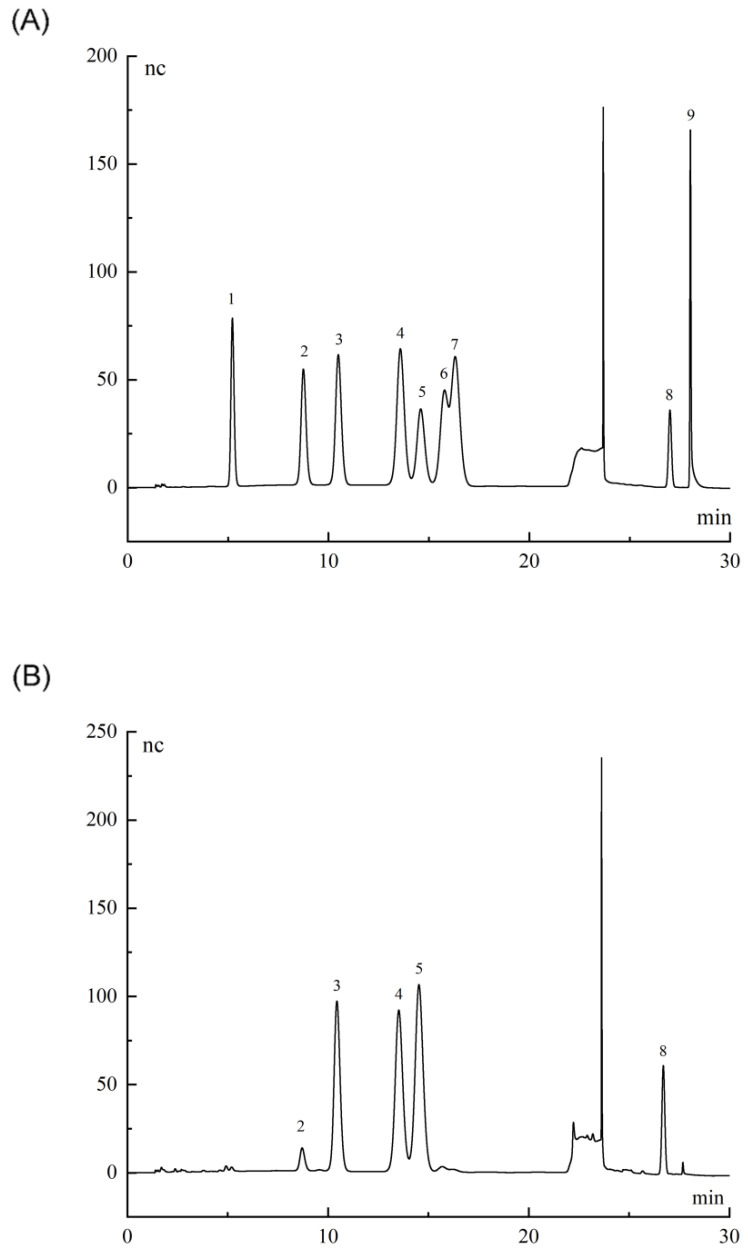
(**A**) Ion chromatogram of standard monosaccharide; (**B**) Ion chromatogram of BCP monosaccharide. (1) Fucose (Fuc); (2) Rhamnose (Rha); (3) Arabinose (Ara); (4) Galactose (Gal); (5) Glucose (Glu); (6) Xylose (Xyl);(7) Mannose (Man); (8) Galacturonic acid (GalA); (9) Glucuronic acid (GluA).

**Figure 7 polymers-14-01119-f007:**
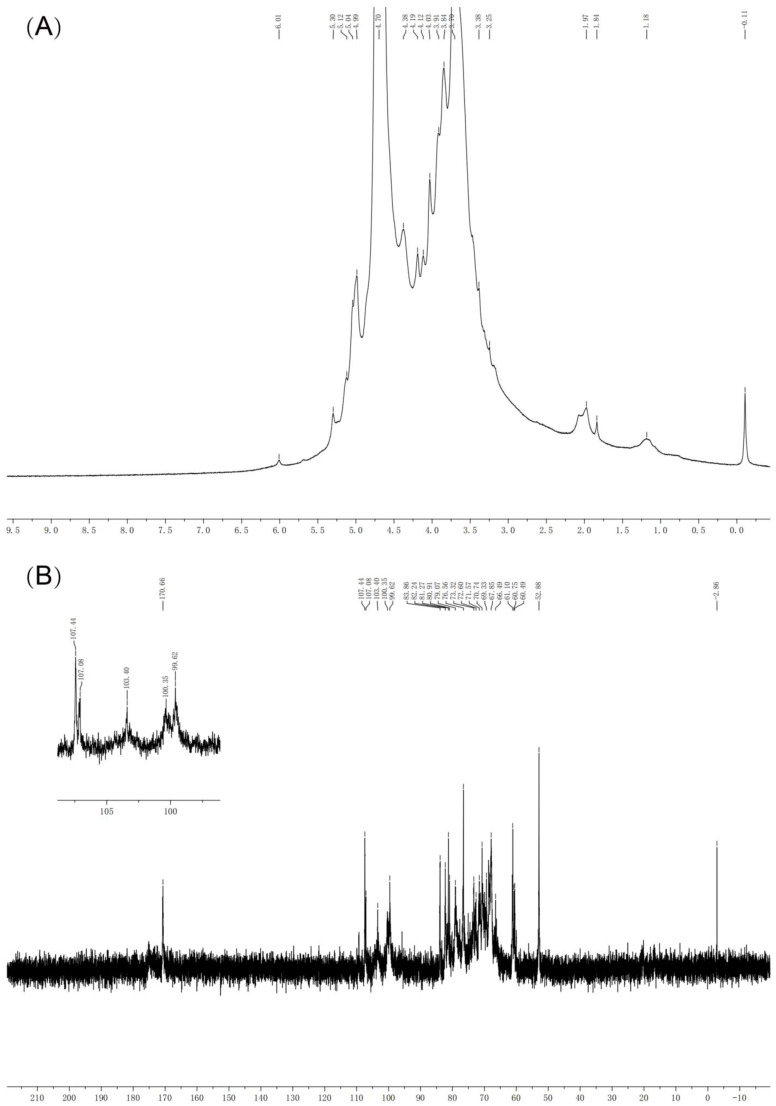
The 1D NMR spectrum of BCP. (**A**) ^1^H spectrum; (**B**) ^13^C spectrum.

**Figure 8 polymers-14-01119-f008:**
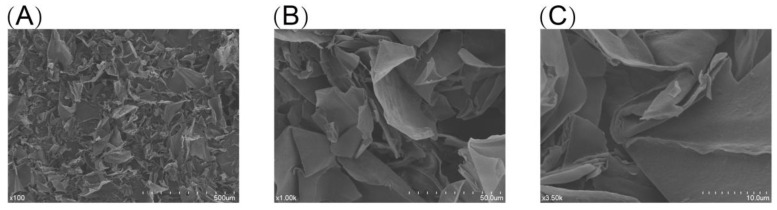
The scanning electron micrographs images of BCP. (**A**) 100×; (**B**) 1000×; (**C**) 3500×.

**Figure 9 polymers-14-01119-f009:**
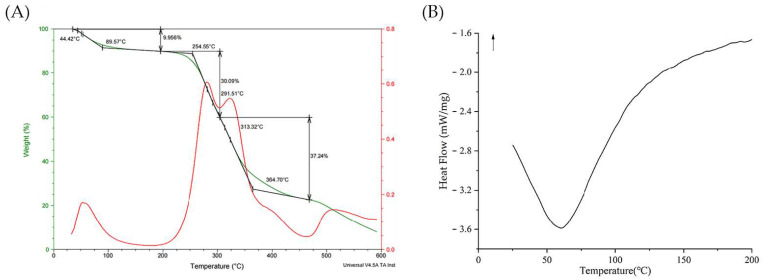
The thermogram curves of BCP. (**A**) TGA; (**B**) DSC.

**Figure 10 polymers-14-01119-f010:**
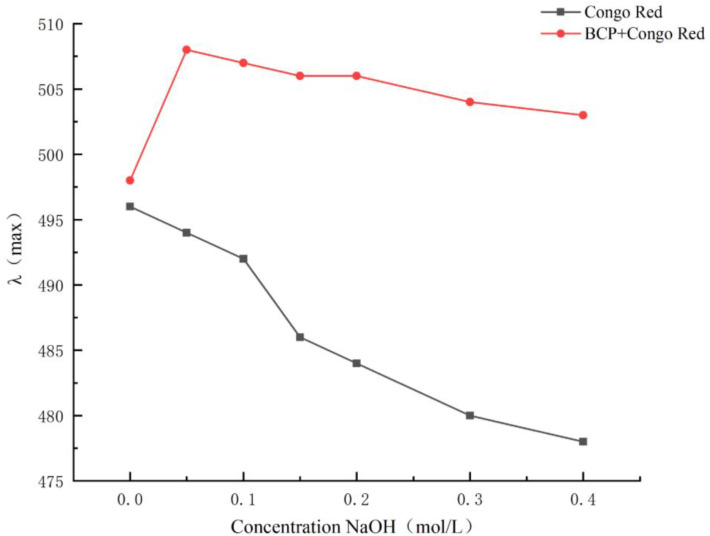
The maximum absorption wavelengths of Congo red and BCP + Congo red at various concentrations of sodium NaOH solution.

**Figure 11 polymers-14-01119-f011:**
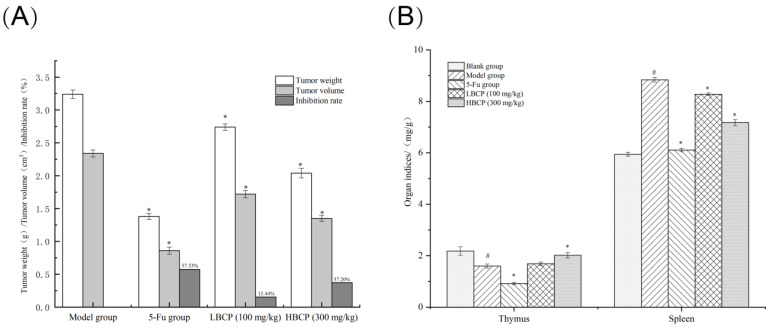
(**A**) Tumor weight and tumor volume of H22 tumor-bearing mice; (**B**) Immune organ indices (Thymus and spleen) of H22 tumor-bearing mice. ^#^
*p* < 0.05 indicated significant compared to blank group; * *p* < 0.05 indicated significant compared to model group.

**Figure 12 polymers-14-01119-f012:**
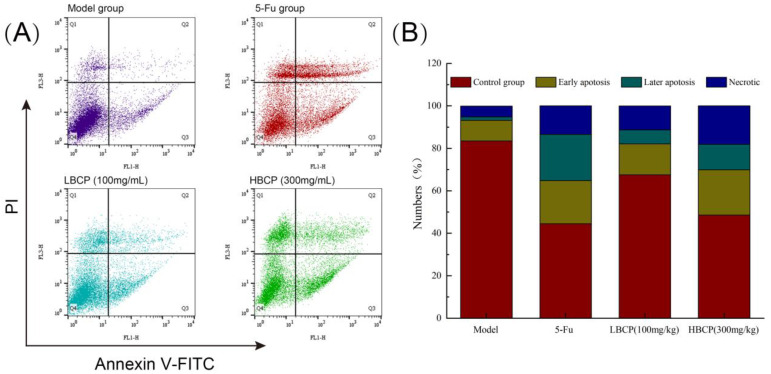
(**A**) Apoptosis of H22 solid tumor cells after FITC-AnnexinV and PI double staining in different groups; (**B**) Bar chart of apoptosis percentage.

**Figure 13 polymers-14-01119-f013:**
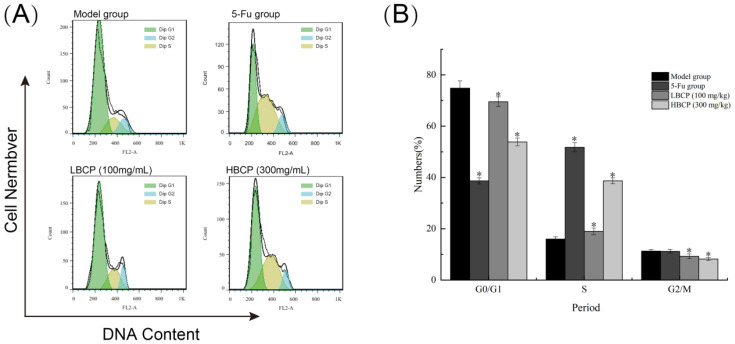
(**A**) Cell cycle distribution of H22 solid tumor cells in mice of different groups; (**B**) Bar graph of the cell population of the corresponding cell cycle phase (G0/G1, S, and G2/M phases). * *p* < 0.05 indicated significant compared to model group.

**Figure 14 polymers-14-01119-f014:**
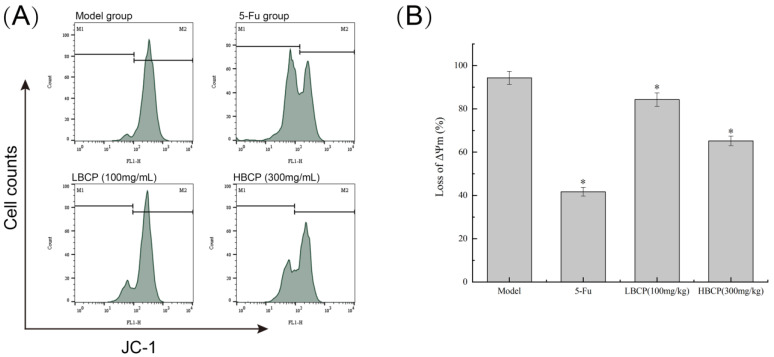
(**A**) MMP disruption in H22 solid tumor cells by JC-1 staining in different groups; (**B**) Bar graph of the mitochondrial membrane potential changes in H22 cells. * *p* < 0.05 indicated significant compared to model group.

**Table 1 polymers-14-01119-t001:** The survival rate of mice.

Group	Blank	Model	5-Fu	LBCP	HBCP
Death Date (d)	-	20	12, 18	9	-
Number of deaths	0	1	2	1	0
Survival Rate (%)	100	90	80	90	100

**Table 2 polymers-14-01119-t002:** Effects of BCP on the body weight.

Group	Dose (mg/kg)	Body Weight (g)				Number of Mice Start/End
		0	7 Days	14 Days	21 Days	
Blank	0	21.98 ± 0.63	26.89 ± 0.69	30.06 ± 0.39	33.10 ± 0.35	10/10
Model	0	22.02 ± 0.28	27.11 ± 0.53	29.17 ± 0.30 ^#^	30.54 ± 0.48 ^#^	10/9
5-Fu	20	21.91 ± 0.45	26.97 ± 1.76	26.49 ± 0.51 *	25.20 ± 0.33 *	10/8
LBCP	100	21.90 ± 0.48	26.59 ± 0.82	29.82 ± 1.67	31.97 ± 0.54 *	10/9
HBCP	300	22.06 ± 0.63	26.68 ± 0.42	29.36 ± 1.19	31.05 ± 0.57 *	10/10

^#^ *p* < 0.05 compared to blank group; * *p* < 0.05 compared to model group.

## Data Availability

All data is available in the manuscript or upon request to the corresponding author.

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
