# Peer review of "The Structural Characteristics of an Acidic Water-Soluble Polysaccharide from Bupleurum chinense DC and Its In Vivo Anti-Tumor Activity on H22 Tumor-Bearing Mice"

_polymers, 2022, doi:10.3390/polym14061119_

Round 1

Reviewer 1 Report

This work is devoted to the study of polysaccharides isolated from Bupleurum chinense, as well as to the study of biological activity. The work is relevant and interesting, since polysaccharides are bioactive components of plants. However, there are a few important points that need to be corrected:
1. In the part of polysaccharide isolation, add experimental data. Drawing conclusions without variations in temperature and duration is too hasty. This part requires detailed information. How does temperature and duration affect monosaccharide composition and molecular weight distribution?
2. It is known that UV analysis is ineffective in the study of polysaccharides. For what purpose did the authors use this method? In this part, either more description with references to the literature is needed, or some other refinement.
3. The GPC data should also be described in more detail. More experimental data is also needed here, as well as more description. This part can be quoted: 10.3390/molecules27010266.
4. IR spectroscopy. Please add functional group designations to the drawing.
5. "3.3. Monosaccharide Composition Analysis". Add more comparison with literary sources. How do your polysaccharides differ from analogues?
6. "The Molecular Morphology". This part needs more description and additions. Please compare your data with those in the literature. Comparison by criteria: size, shape, morphology, etc. You can compare with the studied polysaccharides: 10.17516/1998-2836-0223, 10.1007/s00226-021-01299-1 and others.
7. Discussion - Are these conclusions or a general generalization of a different nature?

Reviewer 2 Report

The authors have reported the extraction method of water soluble acidic polysaccharide (BCP) and showed the antitumor activity in mice. The designing of the method is well described with all physicochemical properties of the functionalized nanoparticles and all the characterization regarding glycoprotein detection. The authors have also described all the physicochemical properties and characterized the sugar contents of the polymers. However, a few points are still missing in this manuscript that needed to be discussed. Therefore, I will recommend the authors to address those points before submitting for final publication.

  • The authors have claimed that the BCP has antitumor activity. How does the structure of polysaccharide play such a significant role?
  • Section 3.7. The description of in vivo experiments is very difficult for the readers to follow. The authors should add a diagram to show how the in vivo experiments were carried out.
  • What is the survival rate of mice?
  • How does the polysaccharide induce apoptosis to cancer cells specifically?
  • Does the polymer internalize the cells? If so, how does the polymer induce mitochondrial membrane damage after internalization?

Round 2

Reviewer 1 Report

The authors significantly improved the quality of the manuscript.

Reviewer 2 Report

The authors have answered all the questions that were asked before. Therefore, I would recommend to send this article for final publications.